# The Role of Axonal Transport in Glaucoma

**DOI:** 10.3390/ijms23073935

**Published:** 2022-04-01

**Authors:** Mariana Santana Dias, Xiaoyue Luo, Vinicius Toledo Ribas, Hilda Petrs-Silva, Jan Christoph Koch

**Affiliations:** 1Intermediate Laboratory of Gene Therapy and Viral Vectors, Carlos Chagas Filho Biophysics Institute, Federal University of Rio de Janeiro, Rio de Janeiro 21941-902, Brazil; mdias@biof.ufrj.br (M.S.D.); hilda@biof.ufrj.br (H.P.-S.); 2Department of Neurology, University Medical Center Göttingen, 37077 Göttingen, Germany; xiaoyue.luo@med.uni-goettingen.de; 3Laboratory of Neurobiology, Department of Morphology, Institute of Biological Sciences, Federal University of Minas Gerais, Belo Horizonte 31270-901, Brazil; ribasvt@ufmg.br

**Keywords:** glaucoma, axonal transport, neurodegeneration, optic nerve head

## Abstract

Glaucoma is a neurodegenerative disease that affects the retinal ganglion cells (RGCs) and leads to progressive vision loss. The first pathological signs can be seen at the optic nerve head (ONH), the structure where RGC axons leave the retina to compose the optic nerve. Besides damage of the axonal cytoskeleton, axonal transport deficits at the ONH have been described as an important feature of glaucoma. Axonal transport is essential for proper neuronal function, including transport of organelles, synaptic components, vesicles, and neurotrophic factors. Impairment of axonal transport has been related to several neurodegenerative conditions. Studies on axonal transport in glaucoma include analysis in different animal models and in humans, and indicate that its failure happens mainly in the ONH and early in disease progression, preceding axonal and somal degeneration. Thus, a better understanding of the role of axonal transport in glaucoma is not only pivotal to decipher disease mechanisms but could also enable early therapies that might prevent irreversible neuronal damage at an early time point. In this review we present the current evidence of axonal transport impairment in glaucomatous neurodegeneration and summarize the methods employed to evaluate transport in this disease.

## 1. Introduction

Glaucoma is a chronic, multifactorial neurodegenerative disease and the major cause of irreversible blindness [1,2]. By 2019, it affected more than 60 million individuals in the world, with predictions of 95.4 million cases by 2030 [2]. The percentage of patients with glaucoma increases dramatically with age, and the increase in life expectancy of the world population drives its prevalence [3,4]. Progressive visual loss affects functional capacity and quality of life, especially in the elderly, and constitutes an important public health problem [5,6,7]. There are several types of glaucoma constituting a multifaceted disease that is less understood than all other eye conditions, as there is still no consensus on its causes and development [1,8]. Glaucoma is characterized by damage followed by progressive loss of RGCs and the retinal nerve fiber layer (NFL), alterations in the optic disc and in the ONH, all of which are diagnosed as a visual field constriction at advanced stage of the disease [1,9]. RGCs are located in the inner part of the retina and have long axons projecting through the optic nerve, which connect the eye to the brain and convey visual information. In the clinic, ONH evaluation, e.g., with fundoscopy or optical coherence tomography (OCT), is crucial to determine the glaucomatous condition. Here, the loss of RGC axons can be visualized and indicates the neurodegenerative process [10]. Glaucoma is the only optic neuropathy known to cause a distinctive ONH “excavated” appearance [8,11]. The pathogenic processes that cause axonal degeneration in glaucoma have been in focus, aiming at a better understanding of the progression of glaucoma and how RGCs degenerate [1,12]. Among structural alterations of RGC axons, the impairment of axonal transport is acknowledged as a factor associated with axonal dysfunction and neurodegeneration in glaucoma [1,13]. Intact axonal transport is essential for proper neuronal function, as it is important for the exchange of molecules, vesicles and organelles across the whole extension of the neuron [14,15]. Impairment of axonal transport has been associated with several neurodegenerative diseases such as Alzheimer’s, Parkinson’s, and Huntington’s disease [16,17,18]. Moreover, several studies already reported transport impairment in axons of RGCs in human glaucoma and different animal glaucoma models [19,20,21,22,23,24].

Here, we describe the evidence that impairment of axonal transport plays an important role in glaucoma, presenting the techniques applied for evaluation of transport and their main outcomes.

## 2. Axonal Damage in Glaucoma

The axons of RGCs form the NFL in the inner part of the retina and converge in the center of the eye, in the optic disc region, to compose the optic nerve. In primates, right after leaving the eye, RGCs axons are non-myelinated and pass through a structure known as lamina cribrosa in the ONH [9,25,26]. The lamina cribrosa is a network mainly composed of collagen fibers, which form pores that allow the passage of the axons to the optic nerve [27]. In rodents, this structure is known as the glial lamina, and is composed mostly of astrocyte projections [28,29,30]. Posterior to the ONH, ensheathment of axons by oligodendrocytes starts in the myelination transition zone (MTZ) and projections proceed to form the myelinated part of the optic nerve [31,32]. Furthermore, astrocytes are major components of the ONH and optic nerve [32]. Another important feature of the ONH is the fact that the RGC axons turn here at an angle of up to 90°, which makes them particularly susceptible for mechanical stress and prone for axonal transport impairment at this location.

In glaucoma, alterations in the ONH region include reduced blood flow, reactive gliosis, oxidative stress, and remodeling of the extracellular matrix [1,23,30,33,34,35]. The mechanism by which RGCs degenerate is, however, not yet clarified. A traditional view is that intraocular pressure (IOP) leads to chronic stress in the ONH, inducing mechanical deformation and physiological alterations in this region. This is considered to be a cause of axonal transport disruptions in glaucoma [36,37,38]. In fact, elevated IOP is a well-known risk factor for the development of glaucoma, and most animal models of glaucoma mimic disease progression by inducing IOP rise [1,12,39]. It should be noted, however, that glaucoma can also develop in normotensive eyes and not every individual with ocular hypertension will develop glaucoma [1,40]. In fact, high IOP itself might not be as important to define glaucoma progression. Short and long term IOP fluctuations have been proposed to impact glaucoma progression [41,42] and orbital cerebrospinal fluid (CSF) pressure may also impact glaucoma development, besides IOP [43,44]. Yet, a great amount of evidence indicates that glaucoma is a vascular optic neuropathy, where reduced blood flow to the ONH occurs as a consequence of alterations in ocular perfusion pressure (OPP), which is defined as the difference between arterial blood pressure and IOP [38,45,46,47]. In this case, a person with high IOP will most likely have a reduced OPP, but not necessarily, and individuals with normal IOP can still have a reduced OPP. Alterations in blood pressure and OPP also seem to correlate with glaucoma progression [46,48,49,50]. Furthermore, it is proposed that glaucoma may be related to a reduced capacity of an individual to adapt ocular blood flow, via autoregulation mechanisms, in response to changes in perfusion pressure [51]. Alterations in blood flow in the eye could impact the supply of nutrients, diminish energy availability and compromise ATP-dependent processes, including axonal transport [38].

Early studies in non-human primates found that elevations in IOP and reduced OPP were associated with obstruction of axonal transport in the ONH [19,33,52,53,54]. At that time, the IOP rise was mostly acute, and in some cases reaching values above 60 mm Hg or even >100 mm Hg, leading to an expressive decrease in perfusion pressure [19,33,52,55,56,57]. Following studies reinforced evidence of axonal transport deficits in glaucoma by using different glaucoma models, including chronic models with moderate IOP elevation (below 30 mm Hg) such as DBA2/J mice and obstruction of aqueous flow by injection of microbeads into the anterior chamber, and several techniques for the evaluation of transport impairment, as we present in this review [22,58,59,60]. Such description is important for understanding disease mechanisms, and indicates that axonal transport deficit is an early event that can compromise the delivery of the organelles and proteins necessary for the correct functioning of axons and favor neurodegeneration (Figure 1).

## 3. Axonal Transport and Neurodegeneration

Intact axonal transport is essential for proper axonal function [14,15]. Intracellular trafficking mechanisms of different elements, such as organelles, proteins, and lipids, are important for the maintenance of any cell type [61,62]. Neurons have a particularly polarized cell morphology that makes transport even more important, with several projections, including dendrites and a long axon, which extend far from the cell body, where most cell machinery is located [14,63]. In particular, the axons of RGCs span a considerable length running from the retina to the superior colliculus and the lateral geniculate nucleus at the posterior part of the brain [64,65]. Axonal transport is an ATP-dependent process, during which cytoskeleton-associated motor proteins actively carry different types of cargos along microtubules in the axons [15]. Transport occurs either in anterograde direction, from soma to axon terminal, or retrograde direction, from terminal to soma. Different directions of movement are associated with a different motor machinery. Anterograde transport is mediated by the kinesin superfamily of motor proteins, while the motor protein complex dynein is utilized in retrograde transport [18,66,67]. The motor proteins move along the cytoskeletal structures of the axon, mainly attached to the microtubules, and their active movement is ATP-dependent [68,69,70]. Furthermore, axonal transport is often divided into fast and slow components (>0.5 vs. <0.1 µm/s), representing high-velocity transport of vesicles and organelles and low-velocity transport of soluble material, such as proteins and polymers, respectively [15]. Important cargos transported anterogradely include synaptic vesicles, pre-synaptic proteins synthesized in the cell body and non-synaptic molecules important for axonal function and structure, such as cytoskeleton and metabolism-related components [15,71,72]. The retrograde axonal transport comprises synaptic components targeted for degradation, as well as neurotrophic factors, which are essential for neuronal survival [15,71,73]. Mitochondria, vesicles, and endosomes may be transported in both directions, while autophagosomes are mostly formed in the axonal tip and undergo retrograde transport to the soma, where the degradation of their contents is finalized in lysosomes [15,18,74,75,76,77].

In accordance with the important role of axonal transport in neuronal function, genetic mutations of transport proteins can lead to neurodegenerative and neurodevelopmental diseases [16,17]. The dysfunctions of cargo trafficking in axons have also been associated with several neurodegenerative diseases, including Alzheimer’s, Parkinson’s, Huntington’s, and amyotrophic lateral sclerosis (ALS) [78,79,80,81,82]. Reports include the identification of axonal swellings with cargo accumulation, as a consequence of transport disruption, in post mortem human samples and altered activity of protein kinases that regulate motor protein function [16,63]. Direct evidence of alterations in axonal transport dynamics have been reported during live imaging of transported cargo in vitro and in vivo, and includes changes such as the number of moving cargos, transport velocity and direction [17,83,84,85,86,87]. Furthermore, proper function of the axonal transport machinery is dependent on cytoskeleton integrity, and modifications of the cytoskeletal structure including microtubule organization and the composition of the actin/spectrin lattice impact axonal transport [88,89]. Cytoskeleton alteration is a common feature of neurodegenerative diseases, including glaucoma [17,90,91]. It is not clear, however, if alterations in axonal transport or the cytoskeleton are the primary event that contributes as a cause of axonal degeneration, or are a consequence of this process [16]. Most probably both scenarios are possible depending on the pathological context.

RGCs present some particular features which could make them even more prone to axonal transport disruptions. They are projection neurons with lengthy axons, estimated to reach 50–100 mm in primates, relying therefore on long-distance axonal transport [65]. Furthermore, the inner retina, where those cells are located, is known to have a high energy demand, and the ONH has normally an increased mitochondria density, a sign of more energy requirement. Since the initial portion of their axons in the NFL and ONH is still non-myelinated, it requires more energy for conduction of action potentials [92,93]. In this sense, RGC axonal transport might be very sensitive to reductions in energy availability. Moreover, RGC axons present an unusual anatomy, as they bend up to 90° to leave the eye to form the optic nerve, which could add up as a physical challenge to axonal transport [65]. When leaving the eye, axons pass through the porous structure of the lamina and face a pressure gradient in the ONH region, between the intraocular and CSF pressures, which in the human eye with a normal IOP is estimated to be 3.5 mm Hg, potentially a further stressor [93]. In this sense, it is not surprising that stress conditions in those axons could rapidly lead to an impairment of their axonal transport. However, the exact mechanism that leads to axonal transport deficits in glaucoma is not clear, and could involve lower blood perfusion and diminished energy availability, glial reactivity, cytoskeleton alterations, molecular triggers or mechanical axonal constriction [59,91,93,94,95,96,97].

## 4. Common Methods to Evaluate Transport Blockage in High IOP

A few recurrent techniques have so far been used to evaluate axonal transport changes as a consequence of high IOP. They are mostly based on the analysis in fixed tissues, which depicts a static scenario at a specific, pre-determined, time point. When there is a blockage in transport, this obstruction in axonal flow leads to an accumulation of transported molecules in the affected region [79]. Thus, a common way to evaluate axonal transport deficits is to detect axonal swellings and/or aggregation of transported material [19,20,23,98]. An alternative approach is to deliver an exogenous, actively transported, substance to one end of the axon, either the cell body or axonal terminal, and evaluate its ability to reach the opposite end of the axon [14]. In this case, once there is axonal blockage, a molecule that is normally transported anterogradely and which was injected into the eye, cannot be identified anymore at the distal end of the RGC axons in the brain [22]. In the same sense, a substance that would normally undergo retrograde transport will not be identified in RGC cell bodies after injection in projection targets in the brain [99].

The first studies of axonal transport impairment after IOP rise used two main techniques of analysis: (i) light and electron microscopy, mostly of the ONH region, looking at axonal swellings and vesicle/organelle accumulations [19,20,33,55,98,100,101]; (ii) intravitreal injections of radiolabeled material, most commonly tritiated leucine, which is incorporated in RGC cell bodies and used for protein synthesis [19,33,52,53,55,100,102]. In this approach, transport blockage could be revealed after autoradiography either by accumulation of radiolabeled material in the ONH, or its lack in projection targets of RGCs in the brain, commonly the lateral geniculate nucleus (LGN) and superior colliculus (SC).

Today, a common way to evaluate axonal transport in high IOP models is through the injection of actively transported molecules either in the eye or in the brain. Among those, intraocular injection of fluorescent conjugates of cholera toxin subunit B (CTB) is used most widely [22,58,59,60,96,103,104,105]. Under normal transport conditions, CTB is internalized by RGCs in the retina after injection in the eye and carried along the axons by active anterograde transport, until it reaches axonal terminals in the brain, especially in the LGN and SC [106,107]. Retrograde transport is commonly evaluated by injection of Fluoro-Gold (FG) in the SC and evaluation of its transport to the RGC cell bodies in the retina [95,99,103,108,109,110]. However, this analysis alone does not discriminate axonal transport and axonal degeneration, since lack of labeling might be due to lost axonal integrity. Moreover, these techniques do not allow a good temporal resolution since the transport of these substances from one end of the RGC to the other takes several hours to days.

Axonal transport blockage can also be analyzed by accumulations of endogenous actively transported material, identified by immunolabeling in tissue sections. Those include, among others, amyloid precursor protein (APP) [23,111], synaptophysin [23,112], BDNF and its receptor TrkB [23,113], IL-6 [114], and also dynein [115].

Recently, new techniques have been explored in order to evaluate axonal transport alterations in live tissues or animals. They include live imaging of labeled mitochondria in the optic nerve ex vivo, in explants of the eye attached to the initial portion of the optic nerve [84,116]. In the living animal, axonal transport has been analyzed through multiphoton live imaging of anesthetized mouse RGCs through the sclera, evaluating movements of labeled mitochondria in the retinal portion of RGC axons [83] or, alternatively, with manganese-enhanced magnetic resonance imaging after intraocular injection of Mn^2+^, which is internalized by neurons via voltage-gated Ca^2+^ channels and carried through axons [117]. These techniques allow a very good temporal resolution, but they are technically quite challenging.

## 5. Evidence of Axonal Transport Impairment in Glaucoma

Several studies investigated axonal transport impairment in glaucoma, as summarized in Table 1, including the appliance of all techniques mentioned above, and the use of a great diversity of animal models with IOP elevation.

### 5.1. First Indications of Axonal Transport Blockage with High IOP

The first indications of axonal transport impairment after high IOP came from studies carried out in the 1970s and 1980s. Most of them used monkeys and induced an increase in IOP by cannulation of the anterior chamber. In this model, a needle is inserted into the anterior chamber of the eye while connected to a saline reservoir, allowing control of IOP according to the height of this reservoir. It is also possible to monitor arterial blood pressure such as to modify IOP to reach a specific perfusion pressure [19]. These reports demonstrated and agreed that axonal transport blockage was present mostly in the ONH region, pointing out the importance of this structure in axonal disfunction. Alterations were identified already in the first hours after IOP rise, indicating that transport deficits might be an early response. Transport blockage included descriptions of axonal swelling with accumulation of mitochondria and other organelles, besides vesicles and multilaminar membranous structures at the ONH [33,100,101], as seen in light and electron microscopy as early as 1 h of high IOP [100]. Furthermore, evaluation of anterograde transport by intraocular injection of radiolabeled amino acids revealed accumulations of proteins in the same region, and decreased labeling of projection targets in the brain [19,52,53,55,100]. Impairment of retrograde transport was identified after injection of horseradish peroxidase into the optic tract or LGN, which also accumulated in the ONH, indicating that transport was compromised in both directions and corroborating the location of blockage [52]. One report by Anderson and Hendrickson identified that transport impairment was worse with more severe increases in IOP, highlighting the correlation between both events [19]. Interestingly, Quigley and Anderson did not detect transport blockage in animals in which IOP was raised for 4 h followed by 4 h of normal IOP, providing initial evidence that blockage can be reverted after normalization of IOP [100]. These studies in the model of anterior chamber cannulation provided important signs of the role of axonal transport in glaucoma, with evidence of location, time, direction and reversibility of blockage.

A few initial reports on transport blockage also used less acute models in monkeys, induced either by laser photocoagulation of the trabecular meshwork or injections of autologous, fixed red blood cells in the anterior chamber. Like the previous model, transport alterations were evident in the ONH and varied with the degree and duration of high IOP [20,55]. Participation of axonal transport impairment in glaucoma was reinforced by evidence of its blockage in the ONH using high IOP models in several other species, including rabbits, pigs and rodents, and also in human glaucoma [98,102,121,122,123]. Importantly, tissue alterations and the accumulation of axonal material in the ONH in human glaucoma specimens observed with light and electron microscopy resembled those of induced IOP elevations in primates [98,122]. The studies which followed in other animal models, as described below, corroborated that axonal transport impairment is an important feature of glaucoma, and was important to its further characterization.

### 5.2. Further Evidence of Axonal Transport Impairment at the ONH in Glaucoma

A great number of reports supported the initial evidence that axonal transport impairment starts in ONH, and showed that it involved several different cargos, as evidenced in a variety of animal models with inducible IOP increase. Accumulations of both BDNF and its receptor TrkB in the ONH region, as identified by immunolabeling, were reported in the acute model by cannulation of the anterior chamber in Brown Norway rats and also after laser photocoagulation of the trabecular meshwork in monkeys, a model that usually leads to pronounced IOP values (>30 mm Hg; Table 1). Furthermore, in the acute model, the presence of radiolabeled BDNF in the retina after SC injection was greatly diminished as compared to control, indicating a reduction in its retrograde transport [113,121]. As discussed above, retrograde transport is important for supplying neuronal cell bodies with neurotrophins secreted by projection targets and internalized in axon terminals. BDNF is an important neurotrophic factor which can be transported complexed with its receptors after binding [133,134]. These studies agreed with the role of the ONH in axonal impairment, and suggested that reduction of BDNF apport to the cell body might be a mechanism by which transport deficits contribute to axonal degeneration.

An important work by Chidlow et al. described the accumulation of several other cargos in the ONH by the immunolabeling of proteins that are normally carried through the transport machinery. Using the glaucoma model by laser photocoagulation of the trabecular meshwork in Sprague–Dawley rats, they identified an increase in APP immunofluorescence in the ONH, which happened as soon as 8 h after induction of IOP rise, peaked at 24 h, persisted at 3 and 7 days, but was barely detected at 14 days after surgery. It was, therefore, an early event in disease progression. Importantly, this increase in APP labeling happened specifically in the non-myelinated part of the optic nerve and did not colocalize with glial markers, pointing out that it was present in axons in the ONH. Accumulation was also seen for the endogenous proteins synaptophysin and BDNF, besides for CTB, that was exogenously provided by intravitreal injection, highlighting the diversity of molecules affected by transport impairment. The author also correlated axonal transport and axonal degeneration, analyzing the latter in transversal sections of the optic nerve stained with toluidine blue. No abnormalities were identified at 1 day after glaucoma induction, when axonal transport deficit was maximal, indicating that transport impairment happens before degeneration [23]. In summary, this work reinforced evidence that axonal transport impairment in ocular hypertension is an early event during glaucoma progression, preceding degeneration, that occurs mostly in the ONH.

APP accumulation in the ONH was also reported to happen early in the microbead model in mice, with moderate IOP increase, detected at 3 days after glaucoma induction [111]. Furthermore, a similar time course of accumulation in ONH axons as observed for APP was reported for IL-6 after laser photocoagulation of the trabecular meshwork. Here, it was identified as early as 8 h after the surgery, and peaked at 1 and 3 days, following a decrease in 7 and 14 days [114]. Other molecules with early axonal accumulation were associated with the glutamatergic presynaptic machinery, such as vesicular glutamate transporter 2 (VGluT2), synaptic vesicle protein 2 (SV2) and synaptophysin, as evidenced 2 days after glaucoma induction by photocoagulation of the limbus and three episcleral veins in CD-1 mice. In this work, retention of these components was associated with the ectopic release of glutamate in the ONH, another mechanism that could contribute to degeneration [112]. Further work conducted by Martin et al. evaluated whether high IOP also impacted the dynein motor complex. There was indeed an accumulation of dynein subunits at the ONH with IOP elevation in Wistar rats at 1 day after 4 h of acute IOP rise by cannulation of the anterior chamber, as well as after laser photocoagulation of the trabecular meshwork, mainly at 3 and 7 days. In this sense, early accumulation of the motor protein dynein was present in the ONH. This provides evidence that transport blockage in glaucoma might not be specific to certain cargos, but reflects a generalized failure, and may broadly affect cell function [115].

Another report of axonal transport impairment in the ONH recently evaluated anterograde axonal transport by intraocular injection of CTB in a chronic glaucoma model with moderate IOP increase, induced by weekly periocular injection of dexamethasone in C57BL/6 J mice. In this case, labeling of CTB diminished completely in the optic nerve posterior to the ONH region and in the SC by 8–10 weeks of high IOP [60]. However, in this work, no early timepoints were evaluated. At these late time points, it is difficult to dissociate functional axonal transport deficits from a general structural damage of the axons.

In summary, a large amount of evidence from glaucoma models with inducible IOP elevation, comprising different durations and magnitudes of IOP increase (depicted in Table 1), concordantly concludes that axonal transport of different molecules is impaired in the ONH early in disease progression.

### 5.3. Characterization of Axonal Transport in DBA/2J Mice

In contrast to the glaucoma models employing induced IOP rise, work in the genetic glaucoma model of DBA/2J mice which naturally develop increased IOP and glaucoma with age, indicates a different spatial and temporal pattern of transport impairment, with evidence that disruption begins distally, in the SC, and progresses distal-to-proximal until it reaches the optic nerve and the retina [22,135,136]. It is important to consider that, since DBA/2J mice develop a more slowly progressive glaucomatous degeneration, among other alterations [135,136], the underlying molecular mechanisms are most probably different from those observed in inducible glaucoma models. Since glaucoma is usually a chronic disease that develops progressively with age, DBA/2J can provide important information about disease processes.

In a detailed work by Crish et al., anterograde axonal transport was evaluated after intraocular injection of fluorescent CTB. In the SC, focal spots of axonal transport deficits first appeared at 8 months of age, identified by areas of SC devoid of label, and progressed with age, until labeling was completely absent in mice with ≥15 months. Therefore, axonal transport impairment was slowly progressive over time. Among animals which did not present SC labeling at 10–12 months, 43% still retained label in optic projections to anterior targets at LGN or olivary pretectal nucleus (OPT) and 29% up to the optic tract, suggesting that transport deficits may first appear in more distal structures. Importantly, RGC presynaptic terminals in the SC maintained their structure, as identified by estrogen-related receptor–β (ERRβ) and VGluT2 labeling, in ages where transport failure had already been identified. Only in the very old animals was the labeling reduced. In the same sense, degenerating axon profiles in the optic nerve composed only 3 to 7% of the total number of axons at 13 months [22]. Therefore, lack of CTB labeling and distal axonopathy was not due to axonal degeneration. In short, axonal transport deficit in this model appeared first in distal RGC projection targets, preceding structural loss, and was progressive.

The fact that axonal dysfunction precedes degeneration in DBA/2J has been confirmed in another work. Here, structural maintenance of SC projections was also observed, with ERRβ labeling preserved in the SC until 12 months of age, after axonal disfunction was identified. Interestingly, anterograde and retrograde transport were not equally affected in glaucomatous mice, indicating that impairment might not be mechanical or due to structural loss in this model. The work combined anterograde (CTB) and retrograde (FG) tracing methods in the same animals and evaluated the time-course of transport deficits in 9–13-month-old DBA/2J. Anterograde transport had a 69% decrease in 9–10 month mice as compared to controls, but retrograde transport presented only a 23% reduction. Therefore, changes in the anterograde direction were detected earlier, and the fact that both directions were differently affected suggests a specific functional deficit of this transport compartment [103].

Smith et al. examined details of the distal axonopathy in DBA/2J mice, investigating the morphology of axon terminals with different degrees of axonal transport deficits, by analyzing the tridimensional ultrastructure of the SC. Projections with transport failure did not have a diminished density of retinal terminals as compared to transport-intact controls, corroborating a maintenance of connectivity. However, CTB– projections had a decreased mitochondrial volume, active zone number and surface area as compared to control [96]. Other alterations have been identified in CTB– projections as compared to CTB+ projections in DBA/2J mice. Transport-deficient CTB– projections in the proximal optic nerve had lower mitochondrial diameter and volume with higher roundness, besides lower axonal length and volume, as well as greater density of autophagic vesicles when compared to CTB+ ones [105]. This evidence combined indicates that transport-deficient projections might present several functional alterations. In the same sense, anterograde transport deficiency of CTB to the SC has been associated with cytoskeleton alterations, including higher levels of pNF-H in the SC and retina, decrease in β-tubulin in the optic nerve and higher amyloid-β42 in the SC than CTB+ projections [91], showing that cytoskeleton alterations are also associated to transport disfunction.

### 5.4. Axonal Transport Impairment and RGC Degeneration

Besides investigations of the relationship between axonal transport and structural maintenance of projections, a few works explored its correlation with RGC degeneration in the retina. In DBA/2J mice, the density of retrogradely labeled FG+ cells in the retina after SC injection was significantly reduced in retinas of 13- and 14-month-old animals, even though the number of NeuN+ RGCs was reduced only at 18 months of age. This indicates that reduced RGC labeling by retrograde transport of FG preceded soma degeneration. Co-injection of FG and DiI, which can undergo passive diffusion in axons, revealed a higher number of DiI+ cells than FG+ in the retina of 13-month-old mice, suggesting the presence of structurally preserved RGCs in which DiI diffuse, but transport of FG is already impaired [99]. Yet, in another investigation, 9-month-old DBA/2J retinas had regions with a reduced or absent retrograde FG label, but that still presented γ-synuclein expression, as opposed to uniform labeling of both markers at 3 months, indicating RGCs that did not degenerate yet, but had impaired transport [108]. These data combined suggest that axonal transport dysfunction precedes RGC degeneration in DBA/2J, emphasizing its initial role in glaucoma progression.

Salinas-Navarro et al. also examined the relationship of active retrograde transport and RGC integrity in the model of laser photocoagulation of the trabecular meshwork in Sprague–Dawley rats. In this work, active retrograde transport was assessed by FG injection in SC. Surviving RGCs in the retina were identified by either pre-label with FG, injected before lesion, or Brn3a immunostaining. Passive axonal diffusion, by dextran tetramethylrhodamine (DTMR) applied to the optic nerve. After ocular hypertension, a lack of retrograde labeling of RGCs by postlesion FG injection appeared already at 8 days, peaking at 8–14 days. Brn3a labeled RGCs were in a greater number (almost 2×) than FG+ RGCs at 8 days, but not at 21 days. This indicates that impairment of retrograde transport happens already at early time points, while cell degeneration appears later. Furthermore, at around 2 weeks, retinas labeled with prelesion FG and postlesion DTMR had a great number of surviving FG+ RGCs but a smaller number of retrogradely labeled DTMR+ RGCs. Since DTMR undergoes passive diffusion, this suggests that there is some physical barrier for it in the axons. When labeling with FG was after lesion, there was a larger retinal area with DTMR+ RGCs than FG+ RGCs at 2 weeks after lesion, indicating, yet, that passive diffusion was more preserved than active retrograde transport from the SC, highlighting that there is also a functional deficit [95]. Overall, the data from this model corroborate an early impairment in axonal transport, before RGC degeneration.

### 5.5. Other Aspects of Axonal Transport Deficits in Glaucoma Models

Other interesting features of axonal transport impairment in glaucoma have been explored. Crish et al. identified that transport deficits were exacerbated with aging. They investigated transport impairment in an inducible glaucoma model, using microbead occlusion of aqueous flow in Brown Norway rats, and identified a decrease in CTB labeling in the SC after 2 weeks of high IOP in aged (7–9 m), but not in young (3–4 m) animals [22]. Another work investigated the influence of fluctuating IOP. After cannulation of the anterior chamber in New Zealand rabbits and rhodamine-β-isothiocyanate (RITC) injection in the eye, Balaratnasingam et al. identified that both fluctuating pressure and sustained high pressure had similar impacts on axonal transport, suggesting that transient changes in pressure also influence transport integrity [129]. An interesting report evaluated the long-term effects of transient IOP elevations, observing no persisting axonal impairment or degeneration after short-term IOP rise. In this work, anterograde and retrograde axonal transport were evaluated by CTB injection in the eye or SC after 1 and 2 weeks of 8 h of high IOP induced by cannulation of the anterior chamber in Brown Norway rats. No obstruction was identified at these later time points, as well as no RGC loss by 6 weeks [104]. This is in line with seminal evidence obtained by Quigley and Anderson [100], indicating that reversal of high IOP may possibly allow the axonal transport to recover and prevent RGC degeneration and glaucoma progression.

### 5.6. Live Imaging of Axonal Transport in Glaucoma

Besides the great amount of evidence obtained in fixed tissue sections that point to an important role of axonal transport disruption in glaucoma, detailed description in living animals is still incipient. Since axonal transport is highly dynamic, real-time imaging is essential for understanding alterations in this process in a reasonable temporal resolution. Live imaging of mitochondrial transport after high IOP has been assessed with two different approaches, providing details of its disruption. Using ex vivo imaging, bidirectional mitochondrial transport in the optic nerve was investigated in transgenic mice expressing CFP in all mitochondria, in explants of the eye attached to the initial portion of the optic nerve. After 1 h of acute IOP rise by ex vivo cannulation of the anterior chamber, there was a 55% decrease in the percent of mitochondria in anterograde motion, indicating a deficit in this transport compartment [116]. Similarly, in a following study, explants from both young (4 m) and older (14–17 m) animals were evaluated after 14 h, 1 or 3 days of the microbead chronic glaucoma model in vivo. After IOP elevation, loss of mitochondria movements was higher in older than younger mice. The number of moving mitochondria diminished with glaucoma in young mice, while in control old animals it was already one third of that in young control mice, and further declined in glaucoma. Furthermore, in young animals the speed of movement increased at 14 h of elevated IOP at both anterograde and retrograde directions, declining afterwards. In older mice, however, almost no mitochondria were moving after 14 h, 1 or 3 days of high IOP [84]. This indicates that both high IOP and ageing can contribute to disruption of mitochondria transport in the optic nerve. Even though this technique allows for the recording of mitochondria movements in the optic nerve, it is important to consider that in this context RGCs lack an active blood flow, altering energy supply, and axons have been axotomized, which per se changes intra-axonal signaling and triggers degeneration [137,138].

Mitochondrial transport in RGC axons has also been studied in vivo, in anesthetized mice, using minimally invasive intravital multiphoton imaging through the sclera in mutant mice with fluorescently tagged mitochondria. This technique allowed for the identification of changes in mitochondria transport after IOP elevations. The authors reported highly dynamic mitochondrial movements in RGC axons under physiological conditions. At 3 days after glaucoma induction by photocoagulation of limbal and episcleral veins in adult mice (4 m), before axonal degeneration is expected, there was a decrease in the number of moving mitochondria in both the anterograde and retrograde direction. There was no alteration in parameters such as duration and distance of transport, duty cycle, and transport velocity. When glaucoma was induced in old (23–25 m) mice, there was a strong decrease in the number of moving mitochondria when compared to adult hypertensive mice, even though those numbers were similar when comparing adult and old controls. This suggests that high IOP impacts mitochondrial transport in RGC axons, with a higher vulnerability in old animals [83]. In this study, though, we should notice that imaging was in the retinal portion of RGC axons, which is anterior to where initial axon damage has been so far reported to occur in glaucoma. Moreover, axons were evaluated at only one time point (3 days) after induction of ocular hypertension, and the transport of other cargos was not examined, neither was the detailed temporal pattern of axonal transport impairment correlated with axonal degeneration analyzed. Overall, both reports on real-time imaging of mitochondria showed decreased transport influenced both by high IOP and age, reconfirming both as important risk factors for glaucoma. They also agree that this is an early feature of glaucomatous degeneration. These alterations in mitochondrial movements may influence energy support to different RGC compartments and contribute to axonal degeneration. However, further studies are required to understand the exact kinetics of axonal transport changes after IOP elevation.

## 6. Neuroprotective Strategies for Glaucomatous Axonopathy

The overview of studies that investigated the role of axonal transport in glaucoma highlighted that its early dysfunction is an important feature of axonopathy in this disease, preceding neurodegeneration. Therefore, improving axonal transport is a promising neuroprotective strategy for glaucoma. Indeed, different experimental strategies with distinct mechanisms of action both enhanced axonal transport and led to neuroprotection in glaucoma [139,140,141,142,143,144]. After an IOP rise to ischemic levels in rats, intravitreal injections of the neurotrophic factor BDNF enhanced anterograde and retrograde transport and increased the levels of the motor proteins dynein and kinesin in the optic nerve. This was in parallel to a neuroprotective role of BDNF, also previously described in several models of RGC lesion [145,146,147,148,149]. In line with these studies, the oral delivery of a synthetic anti-inflammatory drug, the sterol derivative HE3286, was able to prevent impairment of anterograde transport in a microbead rat model and ameliorated several aspects of the disease, leading to enhanced levels of BDNF in the retina and the ONH [150]. This corroborates the important role of neurotrophic factor depletion for RGC degeneration. In another study, overexpression of a modified erythropoietin, a cytokine that enhances red blood cell production, preserved anterograde transport and axonal integrity, possibly by reducing oxidative stress [143,151]. These studies indicate that axonal transport preservation is associated to neuroprotection, and should be a focus of therapeutic strategies to glaucoma.

Some neuroprotective targets for glaucoma are related to the enhancement of energy availability, and likely influence axonal transport. Evidence suggests that reduction of blood flow is a consequence of IOP rise, decreasing energy supply, and could potentially impact axonal transport [152]. Furthermore, metabolic alterations, mitochondrial abnormalities and reduced ATP production are associated with glaucomatous degeneration [153,154,155], indicating that reduced energy availability contributes to disease. Supplying metabolic substrates is protective to axonal degeneration in glaucoma models, as shown after induction of hyperglycemia in rats [156]. NAD+/NADH are cofactors central to several metabolic pathways for ATP production, and also to cell signaling [157,158]. Alterations in NAD metabolism are associated to neurodegenerative diseases, and NAD+ supplementation was neuroprotective in different neurodegenerative contexts, including models of Parkinson’s disease, Alzheimer’s disease, and glaucoma [158,159,160]. Nicotinamide, a precursor of NAD+, was protective in the glaucoma model of microbead injection and after optic nerve axotomy [155,161,162]. Similarly, overexpression of nicotinamide mononucleotide adenylyltransferase (NMNAT) isoforms, an enzyme that catalyzes the last step of NAD+ synthesis from nicotinamide, strongly protected RGC axons in glaucoma models [161,163]. Thus, modulating NAD metabolism is an effective strategy to prevent RGC degeneration in glaucoma, and a possible benefit is enhanced axonal transport due to higher energy availability.

Other neuroprotective strategies for axonopathies can prevent axonal degeneration in glaucoma models. Wallerian degeneration Slow (WldS) mice were discovered as a spontaneously mutated substrain of C57BL/6 that was resistant to Wallerian degeneration after axonal transection, an active process of granular axonal disintegration that occurs distal to a lesion site [160,164,165]. WldS mice animals express a mutant chimeric protein containing 70 amino acids of the N-terminal region of ubiquitination factor E4B (Ube4b) fused to nicotinamide mononucleotide adenylyltransferase 1 (NMNAT1) [166]. This protein replaces axonal NMNAT2, a short-lived protein isoform that is constantly supplied to axons by axonal transport, and is therefore depleted once transport is impaired [167,168]. This phenotype is protective in several neurodegenerative diseases with Wallerian-like degeneration, including glaucoma, in which it led to improved axonal transport of CTB to the SC [34,160,161,169]. Interestingly, administration of nicotinamide in WLDs DBA/2J mice enhanced axonal protection, showing that this combination is a potent neuroprotective strategy to glaucoma [161]. The WldS phenotype is mimicked after loss-of-function mutations in the gene for sterile alpha and TIR motif containing 1 (SARM1), a toll-like receptor adaptor protein with NADase activity [170,171,172]. SARM1 is activated after depletion of axoplasmic NMNAT2 [173], leading to NAD+ depletion in axons and degeneration [170]. SARM1 deficiency reduces axonal degeneration in several disease models, including neuroinflammatory glaucoma and optic nerve crush [170,174,175,176]. These mechanisms are central to axonal disfunction and degeneration, comprising important targets for axon protection in glaucoma.

Additionally, aiming at the cytoskeleton can improve transport in neurodegenerative diseases [17,177]. Cytoskeleton alterations have been associated with glaucoma and are potentially related to transport impairment [23,91,93,178]. Microtubule stabilizing drugs improved axonal transport and attenuated neurodegeneration in mouse models of ALS, Alzheimer’s and Parkinson’s disease [179,180,181,182]. Furthermore, microtubules are regulated by different types of post-translational modifications [183,184] and acetylation of tubulin can influence transport and reduce the recruitment of motor proteins [185,186]. Inhibition or genetic deletion of histone deacetylase 6 (HDAC6), an enzyme that, among other functions, removes acetyl groups from microtubules, improved axonal transport and neurodegeneration in models of neurodegenerative diseases such as ALS, Huntington’s and Parkinson’s [185,186,187,188,189]. Interestingly, increased levels of tubulin acetylation were associated with a WldS-like phenotype and could be related to resistance to axonal degeneration [190]. Considering the important role of axonal transport impairment and cytoskeleton alterations in glaucomatous degeneration, those strategies could attenuate axonopathy and be beneficial to glaucoma.

Overall, therapeutic targets that counteract axonal degeneration and improve axonal transport can induce neuroprotection and are promising strategies for the treatment of glaucoma. However, it still remains to be explored whether there are specific mechanisms of axonal transport dysfunction in glaucoma, including particular changes in specific motor proteins, kinases, transport machinery components, cytoskeleton or transport cargos. Therefore, further studies on glaucomatous axonopathy are needed and could identify novel therapeutic targets.

## 7. Conclusions

Axonal transport is important for proper neuronal function and transport disruption is associated to neurodegenerative diseases. In glaucoma, studies indicate that axonal transport impairment happens early in disease progression and is an important feature in axonal damage and neurodegeneration. Published studies mainly comprise time-point analysis in fixed tissues, using either injection of exogenous actively transported molecules or immunolabeling of endogenous molecules that undergo active axonal transport. They comprise several glaucoma animal models and include evidence of impaired axonal transport in human glaucoma. The ONH is characterized as the primary site of axonal transport impairment, at least in most studies. Furthermore, deficits in response to high IOP seem to be exacerbated by ageing, combining two of the major risk factors for the development of glaucoma. However, several controversies still exist, and it is still not clear whether some observations are due to functional transport disruption in the axons, axonal compression, or are a consequence of axonal degeneration. Furthermore, details of changes in axonal transport dynamics in response to high IOP are still missing. In this sense, future work should focus on finer and more precise evaluations of axonal transport alterations in glaucoma models, including a broader application of live imaging for the description of transport dynamics with good temporal resolution, analysis of different cargos, and over the course of disease progression. Understanding axonal transport changes in glaucoma and its molecular basis will trace the path to therapeutic interventions targeting this event, which happens early in disease development.

## Figures and Tables

**Figure 1 ijms-23-03935-f001:**
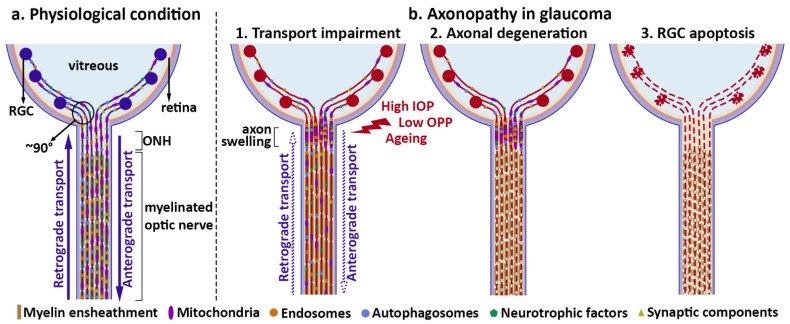
Axonal transport impairment and RGC degeneration in glaucoma. (**a**) Physiological condition, in which axonal transport integrity guarantees the exchange of several molecules, vesicles and organelles throughout the axon, important to maintain RGC function. (**b**) Axonopathy in glaucoma. Here, axonal transport impairment is an early event in disease progression, leading to accumulation of several components in the ONH, that would normally be transported along the axon (1). Alterations in axonal transport are associated with alterations of the cytoskeleton, preceding axonal degeneration (2) and ultimately, apoptosis of RGC cell bodies in the retina (3). Abbreviations: RGC = retinal ganglion cells; ONH = optic nerve head; IOP = intraocular pressure; OPP = ocular perfusion pressure.

**Table 1 ijms-23-03935-t001:** Studies that evaluated axonal transport with ocular hypertension, separated by the main technique used to assess transport impairment.

Reference	Glaucoma Model	Species	IOP	Evaluation of Axonal Transport	Main Outcomes
**Intraocular injection of radiolabeled molecules**					
[19]	Cannulation of the anterior chamber	Owl monkey	15–105 (PP = 108–(–)5 mm Hg)	Intravitreal injection of tritiated leucine; electron microscopy	Accumulation of radioactive labeling in LC with reduced labeling in LGN at 8 h of high IOP. More evident with higher IOP, with axons in the LC dilated and accumulation of mitochondria and vesicles.
[118]	Cyclocryotherapy	Rhesus monkey	20–55 mm Hg	Intravitreal injection of tritiated leucine	Labeling accumulated in lamina scleralis of the ONH after 6, 24 or 48 h of surgery.
[100]	Cannulation of the anterior chamber	Owl monkey	PP = 30 mm Hg	Intravitreal injection of tritiated leucine; electron microcopy	Accumulations in the ONH within 2 h in autoradiography and 1 h in electron microscopy. With 4 h of high IOP + 4 h of normal IOP, no sign of transport blockage.
[52]	Cannulation of the anterior chamber	Cynomolgus monkey	25–150 mm Hg	Intravitreal injection of tritiated leucine; HRP injection into OT or dLGN	Tracers accumulated in lamina scleralis.
[53]	Cannulation of the anterior chamber	Squirrel monkey	20–50 mm Hg; 50–90 mm Hg	Intravitreal injection of tritiated leucine	Accumulation of transported material in ONH, mainly in superior and inferior poles. It was worse for higher IOP.
[55]	Anterior chamber injections of autologous, fixed red blood cells	Squirrel and cynomolgus monkey	24–73 mm Hg	Intravitreal injection of tritiated leucine; electron microscopy	IOP elevation for 2–4 days, 1 week or longer led to accumulated material in ONH, depending on height and duration of IOP elevation. With IOP rise for 2–4 days followed by 1 month of normal IOP, no accumulations were identified, but there were signs of degeneration.
[33]	Cannulation of the anterior chamber	Cynomolgus monkey	PP = 25 (mean IOP 97 mm Hg) or PP = 0	Intravitreal injection of tritiated leucine; electron microscopy	After 4 h of high IOP, accumulation of radiolabeled material and organelle in the ONH. It was the same for animals maintained in a hyperbaric chamber as for room air-breathing ones.
[102]	Cannulation of the anterior chamber	Rabbit	30 or 50 mm Hg	Intravitreal injection of tritiated leucine	Mild accumulation of radiolabeled material in the ONH at 3 h of high IOP.
[119]	Cannulation of the anterior chamber	Japanese monkey	PP = 30 mm Hg (mean IOP ~75 mm Hg)	Intravitreal injection of tritiated leucine or proline	At 5 h of raised IOP, accumulation of radioactive protein in LC with decrease in the optic nerve, especially in its temporal portion.
[120]	Cannulation of the anterior chamber (acute) or laser photocoagulation of the trabecular meshwork	Macaque monkey	Acute: 40–100 mm Hg; LP: 35–48 mm Hg	Intravitreal injection of tritiated leucine	Decrease of labeling in LGN after acute high IOP (12 h) and LP (2–44 weeks). After LP, monkeys had a greater decrease in the magnocellular than in the parvocellular layers of the dLGN.
[121]	Cannulation of the anterior chamber	Brown Norway rat	PP = 25 (mean IOP 50 mm Hg) or PP = 0	Injection of radiolabeled BDNF in SC	Reduction in BDNF transported to the NFL after 6 h of high IOP.
**Light and electron microscopy**					
[20]	Laser treatment of the trabecular meshwork	Rhesus monkey	Mean 26–50 + mm Hg	Light and electron microscopy	Axonal swellings in the ONH (3–11 weeks).
[98]	Human glaucoma	Human	–	Electron microscopy	At scleral lamina, axons were swollen with major obstruction of organelle, including vesicles, mitochondria, and multivesicular bodies. Accumulated material and location were similar to findings of induced high IOP in primates.
[101]	Cannulation of the anterior chamber	Owl monkey	PP = 35 mm Hg	Electron microscopy	Accumulation of membranous organelles, such as mitochondria and microvesicles within axons after 4 h.
[122]	Human glaucoma	Human	–	Light and electron microscopy	Accumulations of organelles in optic nerve axons at the LC.
[123]	Cannulation of the anterior chamber	Guinea Pig	60 mm Hg	Electron microscopy	At 4 h of high IOP, accumulation of organelles and vesicles in axons.
[124]	Human glaucoma	Human	–	Light microscopy	Axonal swellings posterior to the lamina, with amorphous and poorly stained material with few nuclei.
[125]	Microbead model	Mouse	Peak ~22 mm Hg	Electron microscopy	At 1–3 days after IOP elevation, axon swelling and accumulated mitochondria and vesicles in axons at the ONH.
**Injection of exogenous actively transported molecules**					
[126]	Cannulation of the anterior chamber	PVG/Mol hooded rat	50 mm Hg; or 10 min 180 mm Hg + 2 h 15 mm Hg	HRP injection into LGN	Lower absorbance in contralateral retinas after 2 and 4 h of high IOP. No decrease after 10 min of high IOP + 2 h normal IOP.
[127]	Cannulation of the anterior chamber	PVG/Mol hooded rat	35 mm Hg; or 2 h of 50 mm Hg + 2 h of 15 mm Hg	HRP injection into LGN	Decrease in HRP content in retina after increased IOP for 4 h. No decrease after 2 h of high IOP + 2 h normal IOP.
[128]	Cannulation of the anterior chamber	Landrace pigs	40–45 mm Hg	Intravitreal injection of RITC	6 h after IOP increase, reduced RITC labeling in the postlaminar tissue.
[21]	Cannulation of the anterior chamber	Landrace pigs	~40–45 mm Hg	Intravitreal injection of RITC	With 12 h of high IOP, RITC was present mainly at prelaminar and LC, and reduced in postlaminar region. Changes in peripheral nerve bundle were more pronounced and earlier than in central nerve. Alterations in neurofilament proteins happened before axonal transport impairment (3 h).
[99]	DBA/2J	Mouse	–	FG and DiI injection in SC	Density of FG+ cells in the retina decreased in 13-14 m old animals, even though NeuN+ density only decreased at 18 m. In 13 m mice, co-injection of FG and DiI led to higher number of DiI+ than FG+ cells in the retina.
[108]	DBA/2J	Mouse	–	FG injection in SC	Retinas of 9 m old mice had regions with reduced of absent FG labeling, but preserved γ-synuclein expression.
[22]	DBA/2J or microbead	Mouse and Brown Norway rat	DBA2J: peak ~25 mm Hg Microbead: 25–30 mm Hg (sustained)	Intravitreal injection of CTB	DBA2J: decrease in CTB labeling appeared first in the SC, starting at 8 months of age, and progressed distal-to-proximal. Microbead: after 2 weeks of high IOP, decrease of CTB labeling happened in aged (7–8 m) but not young (3–4 m) rats.
[95]	Laser photocoagulation of the trabecular meshwork, perilimbar and episcleral veins	Sprague-Dawley rat	Peak ΔIOP ~ 20 mm Hg	FG and DTMR injection in SC	After 8 days of high IOP, there was a greater number of Brn3a+ cells in the retina then FG+ cells, as labeled post lesion. At 2 weeks after lesion, there was a larger retinal area DTMR+ (passive diffusion) than FG+.
[129]	Cannulation of the anterior chamber	New Zealand White rabbit	40 mm Hg; fluctuation of 7.5 and 57.5 mm Hg	Intravitreal injection of RITC	RITC intensity was diminished in the optic nerve after 6 h of high IOP. A similar decrease was observed in eyes with fluctuating high pressure, in which IOP was changed between 7.5 mm Hg and 57.5 mm Hg at 30 min intervals.
[130]	Translimbal laser photocoagulation (trabecular meshwork + perilimbal veins)	Wistar rat	Peak ~40–50 mm Hg	FG injection in SC	29 days after surgery, retinas contained RGCs that were FG–, but Sncg+ and/or labeled with pNF in somas and dendrites.
[109]	Laser photocoagulation of the trabecular meshwork	Sprague-Dawley rat	34.8 mm Hg (day 1)	FG injection in SC	FG spectrometry identified a reduction in FG levels in the SC after 5 days of high IOP.
[104]	Cannulation of the anterior chamber	Brown Norway rat	50 mm Hg	Intravitreal and SC injection of CTB	No decrease in CTB labeling was observed 1–2 weeks after a reversible IOP increase of 8 h.
[103]	DBA/2J	Mouse	–	Intravitreal injection of CTB + FG injection in SC	CTB labeling in the SC was decreased by 69% in 9–10 m old mice, while FG labeling in the retina only diminished by 23%.
[131]	Injection of cultured conjunctival cells into the anterior chamber	Marshall ferret	Mean 42.8 mm Hg (sustained)	Intravitreal injection of CTB	13 weeks after surgery, there was a great reduction of CTB labeling in SC and LGN.
[58]	Microbead	C57 mouse	~20 mm Hg (sustained)	Intravitreal injection of CTB	After 5 weeks of IOP elevation, diminished CTB labeling in contralateral SC and LGN.
[105]	DBA/2J	Mouse	–	Intravitreal injection of CTB	In 11–14 m old mice, projections in CTB- ONs had decreased axonal volume and length, with greater volume of autophagic vesicles than CTB+ ones. Mitochondria had lower volume and diameter and higher roundness.
[132]	DBA/2J	Mouse	–	Intravitreal injection of CTB	In 12–15 m old mice, IL-6 was elevated in CTB- collicular regions compared to areas with intact transport.
[110]	Cannulation of the anterior chamber	Sprague-Dawley rat	40 mm Hg or PP = 25	Intravitreal injection of RITC + FG injection in SC	With 6 h of high IOP, reduction of RITC label in the ON after 24 h of baseline, and of FG label in the retina after 6 h of baseline.
[96]	DBA/2J	Mouse	–	Intravitreal injection of CTB	Reduced labeling of SC in DBA/2J. CTB- projections did not have less retinal boutons, but had lower mitochondrial volume, active zone number and surface area.
[91]	DBA/2J	Mouse	–	Intravitreal injection of CTB	CTB- projections had higher levels of pNF-H in SC and retina, lower decrease in β-tubulin in the ON and higher amyloid-β42 in the SC than CTB+ ones.
[59]	Weekly injections of chondroitin sulfate into the anterior chamber	Wistar rat	21–23 mm Hg (sustained)	Intravitreal injection of CTB	Reduction of CTB labeling in myelinated ON (6 weeks), besides SC and LGN (6 and 15 weeks).
[60]	Weekly periocular injection of dexamethasone	C57BL/6 J mouse	~16–21 mm Hg (sustained)	Intravitreal injection of CTB	Decrease in CTB labeling in the ON and SC between 8–10 weeks.
**Immunolabeling of endogenous actively transported material**					
[113]	Cannulation of the anterior chamber (acute) or laser photocoagulation of the trabecular meshwork	Brown Norway rats or cynomolgus monkey	Acute: 51–81 mm Hg (PP = 0); 19–58 mm Hg (PP = 25 mm Hg) LP: peak 25–43 mm Hg	Labeling of TrkB and BDNF; radiolabeled BDNF injection into SC; light and electron microscopy	BDNF and/or its receptor accumulated in acute high IOP (4 h) and LP (2 m–2 y). In acute model, there was decreased transport of BDNF to the retina and axons in ONH were swollen, with accumulated vesicles.
[115]	Cannulation of the anterior chamber or laser photocoagulation of the trabecular meshwork	Wistar rat	Acute: PP = 25 mm Hg; LP: peak IOP 37–38 mm Hg	Dynein	There was accumulation of dynein subunits in the ONH at 1 day after 4 h of acute model or after 3–7 days (mainly) of LP.
[23]	Laser photocoagulation of the trabecular meshwork	Sprague-Dawley rat	Peak ΔIOP ~ 26 mm Hg	APP, synaptophysin, BDNF; intravitreal injection of CTB	Protein accumulation in axons at the ONH as soon as 8 h, peaking at 24 h.
[112]	Photocoagulation of limbus (270°–300°) and three episcleral veins	CD-1 mouse	>21 mm Hg	SV2, synaptophysin, VGLUT2, SNAP-25, VAMP2, Bassoon.	Accumulation of components of the glutamatergic presynaptic machinery in ONH at 2 days after surgery.
[114]	Laser photocoagulation of the trabecular meshwork	Sprague-Dawley rat	Peak ΔIOP~25 mm Hg	IL-6	Accumulation of IL6 in ONH axons with 8 h of high IOP, and mainly at 1 and 3 days.
[24]	Injection of hypertonic saline solution into episcleral vein	Brown Norway rat	Peak 35–40 mm Hg	Tau	At 3 weeks of high IOP, tau protein accumulated in the retina and diminished in the ON.
[111]	Microbead	CD1 mouse	Peak ΔIOP~10–15 mm Hg	APP	At 3 days of high IOP, the fraction and the mean intensity of suprathreshold pixels was higher in ONH and ON.
**Imaging in live tissue/animals**					
[83]	Photocoagulation of limbal (300°) and episcleral veins	Thy1-mito.CFP mouse	Peak ~40 mm Hg	Intravital multiphoton imaging of anesthetized mouse RGCs through the sclera	Decrease in the number of moving mitochondria at 3 days of high IOP in adult mice (4 m), which was worse in old (23–25 m) mice.
[116]	Ex vivo cannulation of the anterior chamber (acute) or bead + sodium hyaluronate injection in anterior chamber (chronic)	Thy1-mito.CFP mouse	Acute: 30 mm Hg; chronic: peak 28 mm Hg (14 h)	Live imaging of globe-optic nerve explants	Decrease in the number of moving mitochondria after 1 h of acute or 3 days of chronic IOP increase.
[117]	DBA/2J	Mouse	Peak 18.4 mm Hg	Manganese-enhanced magnetic resonance imaging	Decrease in Mn^2+^ presence in both SC and LGN in 14-month-old DBA/2J mice after intraocular injection.
[84]	Microbead	Thy1-mito.CFP mouse	–	Live imaging of globe-optic nerve explants	Alterations in mitochondria movements after 14 h, 1 or 3 days of high IOP. Loss of mitochondria movements was more severe in old (14–17 m) than in young (4 m) mice.

IOP = intraocular pressure; PP = perfusion pressure; LC = lamina cribrosa; LGN = lateral geniculate nucleus; IOP = intraocular pressure; ONH = optic nerve head; HRP = horseradish peroxidase; dLGN = dorsal lateral geniculate nucleus; SC = superior colliculus; RITC = rhodamine-β-isothiocyanate; FG = Fluoro-Gold; DTMR = dextran tetramethylrhodamine; CTB = cholera toxin subunit B; LP = laser photocoagulation; ON = optic nerve.

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
