# Peer review of "The Role of Axonal Transport in Glaucoma"

_ijms, 2022, doi:10.3390/ijms23073935_

Round 1

Reviewer 1 Report

The authors review the role of axon transport in glaucoma. The review is well-written and covers broad aspects associated with axon transport pathophysiology in the disease. Table 1 is a useful resource. There are a few minor points to address, as described below:

  • There are areas where additional references are required and should be inserted for clarity and information for the reader. For example:
    - line 53 – after glaucoma models, please provide references
    - lines 107-132 cites zero references, this section requires references to point the reader in the direction of important publications that reference axonal transport proteins and general axonal transport mechanisms.
  • Line 137 – after ALS – the authors should provides specific references to these diseases and association with axonal transport
  • Can the authors explain in the text or expound on “sign of synaptic integrity” on line 130, it is not clear for the reader.
  • Line 156 –“could require more energy” - it does require more energy. Be explicit here!
  • Line 157 – “axonal transport of RGCS” reads better as “RGC axonal transport”
  • Line 191 – “nowadays” is very casual language, consider instead “Today”
  • Can the authors consider a column in table 1 for species, it would make it a little clearer to find relevant information and make the table easier to digest.
  • Optic tract on line 246 does not need to be shortened to “OT” as it is not used elsewhere.
  • Line 247, remove “yet”, line 248 “in animals in which”
  • Line 451 “favored by ageing” – I am not sure this is exactly what is meant by the authors. Do you mean “exacerbated by”?
  • Can the authors expand a little on what would encompass “finer and more precise” in the conclusions statements….this statement seems very vague and examples to improve current studies should be given.

Author Response

We appreciate the reviewer’s comments and criticism, and we have incorporated suggestions as delineated below. We trust that these modifications have improved our manuscript and we look forward to learning the disposition for this work.

Reviewer #1

Comments to the Author

The authors review the role of axon transport in glaucoma. The review is well-written and covers broad aspects associated with axon transport pathophysiology in the disease. Table 1 is a useful resource. There are a few minor points to address, as described below:

Specific comments and answers

Comment 1. There are areas where additional references are required and should be inserted for clarity and information for the reader. For example:

- line 53 – after glaucoma models, please provide references

- lines 107-132 cites zero references, this section requires references to point the reader in the direction of important publications that reference axonal transport proteins and general axonal transport mechanisms.

Line 137 – after ALS – the authors should provide specific references to these diseases and association with axonal transport

Answer 1. Thank you for the comment. We have inserted the appropriate references to the sentences indicated, as well as in lines: 28, 33, 35, 41, 42, 44, 48, 50, 67, 87, 138, 141, 143, 156, 187, 189, 191, 194, 196, 211.

All locations with new references are highlighted in yellow.

Comment 2. Can the authors explain in the text or expound on “sign of synaptic integrity” on line 130, it is not clear for the reader.

Answer 2. Thank you for the comment. To keep the sentence clear, the authors decided to remove this part. The whole sentence was changed from “The retrograde axonal transport comprises synaptic components targeted for degradation, as well as neurotrophic factors, a signal of synaptic integrity which is essential for neuronal survival” to “The retrograde axonal transport comprises synaptic components targeted for degradation, as well as neurotrophic factors, which are essential for neuronal survival”

Comment 3. Line 156 –“could require more energy” - it does require more energy. Be explicit here!

Answer 3. Thank you for the comment. We changed the sentence from  “could require more energy” to “require more energy”.

Comment 4. Line 157 – “axonal transport of RGCS” reads better as “RGC axonal transport”

Answer 4. Thank you for the comment. The sentence has been changed as suggested.

Comment 5. Line 191 – “nowadays” is very casual language, consider instead “Today”

Answer 5. Thank you for the comment. The word has been changed as suggested.

Comment 6. Can the authors consider a column in table 1 for species, it would make it a little clearer to find relevant information and make the table easier to digest.

Answer 6. Thank you for the comment. The column “Glaucoma model” was subdivided into “Glaucoma model” and “Species”.

Comment 7. Optic tract on line 246 does not need to be shortened to “OT” as it is not used elsewhere.

Answer 7. Thank you for the comment. It was changed as suggested.

Comment 8. Line 247, remove “yet”, line 248 “in animals in which”

Answer 8. Thank you for the comment. Both changes have been applied.

Comment 9. Line 451 “favored by ageing” – I am not sure this is exactly what is meant by the authors. Do you mean “exacerbated by”?

Answer 9. Thank you for the comment. The sentence has been changed from “Furthermore, deficits in response to high IOP seem to be favored by ageing, combining two of the major risk factors for development of glaucoma.” To “Furthermore, deficits in response to high IOP seem to be exacerbated by ageing, combining two of the major risk factors for development of glaucoma.”

Comment 10. Can the authors expand a little on what would encompass “finer and more precise” in the conclusions statements….this statement seems very vague and examples to improve current studies should be given.

Answer 10. Thank you for the comment. This sentence has been expanded from “In this sense, future work should focus on finer and more precise evaluations of axonal transport alterations in glaucoma models.” to “In this sense, future work should focus on finer and more precise evaluations of axonal transport alterations in glaucoma models, including a broader application of live imaging for description of transport dynamics with good temporal resolution, analysis of different cargoes, and over the course of disease progression.”

Reviewer 2 Report

This is an excellent review by Dias et al. I appreciate the in-depth review of each major article and the wonderful Table provided.

I have 3 major criticism. 

1) Starting at section 5 and beyond, please summarize studies, indicate how they relate to one another, and what the studies mean to the field (i.e., provide synthesis and breadth). 

2) Please provide an illustration to accompany section 2-3. 

3) The author may consider adding a section on neuroprotective targets for axonopathy (i.e., NAD, WldS, Sarm1).   

The remaining critiques are minor:

line 37 - "perceived" maybe changed to diagnosed

line 159 - 90 deg - superscript deg symbol

line 247 - spelling - Quingley tp Quigley

line 334 - please simplify sentence

line 339 - this is a confusing sentence. please simplify

line 417 - spelling - recoding to recording

Author Response

We appreciate the reviewer’s comments and criticism, and we have incorporated suggestions as delineated below. We trust that these modifications have improved our manuscript and we look forward to learning the disposition for this work. 

Reviewer #2

Comments to the Author

This is an excellent review by Dias et al. I appreciate the in-depth review of each major article and the wonderful Table provided.

Specific comments and answers

I have 3 major criticism. 

Comment 1. Starting at section 5 and beyond, please summarize studies, indicate how they relate to one another, and what the studies mean to the field (i.e., provide synthesis and breadth). 

Answer 1. Thank you for the comment. The entire section 5 has been reviewed to accommodate these changes. We trust they are appropriate.

Comment 2. Please provide an illustration to accompany section 2-3. 

Answer 2. Thank you for the comment. An illustration has been added to the paper, below line 107.

Comment 3. The author may consider adding a section on neuroprotective targets for axonopathy (i.e., NAD, WldS, Sarm1).   

Answer 3. Thank you for the comment. A new section named “Neuroprotective strategies for glaucomatous axonopathy” has been added before conclusions.

The remaining critiques are minor:

Comment 5. line 37 - "perceived" maybe changed to diagnosed

Answer 5. Thank you for the comment. The word has been changed as suggested.

Comment 6. line 159 - 90 deg - superscript deg symbol

Answer 6. Thank you for the comment. The change has been applied.

Comment 7. line 247 - spelling - Quingley to Quigley

Answer 7. Thank you for the comment. The name has been changed.

Comment 8. line 334 - please simplify sentence

Answer 8. Thank you for the comment. The sentence has been change from “Smith et al. examined the distal axonopathy in DBA/2J mice through tridimensional ultrastructure of the glaucomatous SC using serial block-face scanning electron microscopy, investigating the morphology of axon terminals in glaucomatous mice with different degrees of axonal transport deficits.” to “Smith et al. examined details of the distal axonopathy in DBA/2J mice, investigating the morphology of axon terminals with different degrees of axonal transport deficits, by analyzing the tridimensional ultrastructure of SC.”

Comment 9. line 339 - this is a confusing sentence. please simplify

Answer 9. Thank you for the comment. The sentence has been changed from “However, CTB- projections did have lower mitochondrial volume, besides decreased active zone number and surface area as compared to control” to “However, CTB- projections had decreased mitochondrial volume, active zone number and surface area as compared to control”.

Comment 10. line 417 - spelling - recoding to recording

Answer 10. Thank you for the comment. The change has been applied.

Round 2

Reviewer 2 Report

Very nice revision!